# Usefulness of Adenosine Deaminase Assay in Diagnosis of Patients with HIV Infection and Pleural Tuberculosis

**DOI:** 10.3390/medsci6040101

**Published:** 2018-11-13

**Authors:** Gabriel Yusti, Mariano Fielli, Alejandra Gonzalez, Graciela Torales, Alejandra Zapata, Adrian Ceccato

**Affiliations:** 1Pneumology Service, Hospital Nacional Alejandro Posadas, El Palomar 1684, Argentina; mhfielli@hotmail.com (M.F.); alestork@yahoo.com.ar (A.G.); aceccato@clinic.cat (A.C.); 2Infectious Diseases Service, Hospital Nacional Alejandro Posadas, El Palomar 1684, Argentina; gratorales@hotmail.com; 3Microbiology Department, Hospital Nacional Alejandro Posadas, El Palomar 1684, Argentina; alejandraz07@hotmail.com

**Keywords:** adenosin deaminase (ADA), pleural tuberculosis

## Abstract

The utility of the adenosine deaminase (ADA) assay in the diagnosis of patients with pleural tuberculosis (TB) and human immunodeficiency virus (HIV) infection is controversial. Forty-eight HIV positive patients with pleural effusion were evaluated; ADA assay was obtained in forty-three of them. Twenty-five patients presented diagnosis of TB. Patients with diagnosis of TB showed a median value of ADA of 70 IU/L (interquartile range (IQR) 41–89) and the non-TB group a median of 27.5 IU/L (IQR 13.5–52). Patients with diagnosis of TB had a median cluster of differentiation 4 (CD4) count of 174 (IQR 86–274) and the non-TB group had a median of 134 (IQR 71–371). Receiver operating characteristic curve was performed with an area under the curve of 0.79. The best cut-off obtained was 35 IU/L with a sensibility of 80% and a specificity of 66%. There was no correlation between CD4 lymphocytes count and the value of ADA in the TB patient group.

## 1. Introduction

There is a high incidence of patients with human immunodeficiency virus (HIV) and *Mycobacterium tuberculosis* co-infection. Out of the 10.4 million people that were diagnosed with tuberculosis (TB) in 2016, 1.4 million (13.4%) were HIV-positive [1]. In 2016, about 0.4 million people died from tuberculosis associated with HIV. That same year, approximately 40% of the deaths recorded in the HIV-positive patients were due to tuberculosis [1]. Pleural TB is the most common cause of extrapulmonary TB [2]. Extrapulmonary TB is more common in HIV-positive patients when compared with HIV seronegative patients [3].

The diagnosis of pleural TB depends on the demonstration of *Mycobacterium tuberculosis* in sputum, pleural fluid, or pleural biopsy specimens. This is often difficult, thus other diagnostic tests have been proposed for the diagnosis of pleural tuberculosis, such as adenosine deaminase (ADA) assay. The ADA test has shown a high sensitivity and specificity in the diagnosis of pleural TB [4], although its usefulness in diagnosis in immunocompromised patients is controversial due to the broad range in results that have been observed across studies [5,6,7].

We aimed to evaluate the usefulness of the adenosine deaminase (ADA) assay in the diagnosis of pleural TB in HIV-infected patients and the usefulness of correlating this value with the cluster of differentiation 4 (CD4) lymphocyte serum count.

## 2. Materials and Methods

We retrospectively reviewed all patients with concomitant presentation of pleural effusion and HIV infection who underwent a diagnostic thoracentesis and ADA level determination in pleural fluid at Hospital Nacional Prof. Alejandro Posadas in Argentina between January 2013 and August 2016.

The pleural fluid samples were centrifuged, the sediment was sown in liquid medium “mycobacteria growth indicator tube” (MGIT), and the supernatant was stored at −20 °C to a dosage of ADA corresponding to the method of Galanti Giusty [8]. The diagnosis of pleural TB was confirmed if culture for *Mycobacterium tuberculosis* in pleural fluid, tissue, or another respiratory sample was positive. The presence of caseous granuloma in the pleural biopsy or an improvement after a tuberculosis treatment were also considered as diagnosis criteria.

The Ethics Committee of the Hospital Nacional Alejandro Posadas approved the study. The study was conducted in accordance with the Declaration of Helsinki 2013, Law 25.326 of the Ministry of Health and the protocol was approved by the ethics committee of Dr. Vicente Federico Del Giùdice (ref: 216 LUPOSO) on 8 October 2018.The need for written informed consent was waived due to the non-interventional nature of this study. Patients’ identity remained anonymous.

Statistical analysis was performed using SPSS (Version 18.0. Chicago: SPSS Inc.) statistical software. Continuous variables were compared using the Student’s *t*-test or Mann-Whitney. Receiver operating characteristic curve was performed to ADA assay and diagnosis of pleural tuberculosis. Linear regression model was performed to evaluate the correlation between ADA level and CD4 lymphocytes serum counts in patients with diagnosis of pleural tuberculosis.

## 3. Results

Forty-three HIV infected patients with pleural effusion were evaluated. Out of the 43 patients, 25 presented diagnosis of TB, 22 of them with positive culture (11 from pleural fluid and 11 from others respiratory samples), and three improved after tuberculosis treatment. In the non-TB group, four patients presented diagnosis of empyema, two of parapneumonic effusions, four of transudate effusions, and three malignant effusions. In five patients, the diagnosis was not achieved, although tuberculosis was ruled out. Patients with TB diagnosis showed a median value of ADA of 70 IU/L (interquartile range (IQR) 41–89) and the non-TB group, 27.5 IU/L (IQR 13.5–52). Patients with diagnosis of TB had a median CD4 count of 174 (IQR 86–274) and the non-TB group, 134 (IQR 71–371) (Figure 1). The mean age in the TB group was 40 (±10) years and 41 (±10) years in the non-TB group; 19 (76%) patients were male in the TB group and 13 (72%) in the non-TB group. Non-differences were observed in fluid characteristics in pH (7.39 ± 0.07 vs. 7.41 ± 0.13), glucose (66 mg/dL ± 27 vs. 83 mg/dL ± 26), proteins (3.95 g/dL ± 1.15 vs. 3.27 g/dL ± 1.47), lactate dehydrogenase (LDH, 882 mmol/L ± 497 vs. 1046 ± 635), and leukocytes (1173/mm^3^ ± 1733 vs. 1375/mm^3^ ± 1024) (Table 1).

Receiver operating characteristic (ROC) curve was performed (Figure 2) with an area under the curve of 0.79 (95% confidence interval (CI) 0.65–0.92).

The best cut-off obtained was 35 IU/L with a sensitivity of 80% (95% CI 59–93) and a specificity of 66% (95% CI 41–86).

In the TB patients group, the correlation between CD4 lymphocytes count and the value of ADA was low (r − 0.12) and without statistical significance.

## 4. Discussion

In this study, the sensitivity and specificity of the ADA assay to diagnose pleural tuberculosis in HIV infected patients were lower than that reported in non-HIV infected patients [9]. There was no relationship between the level of immunodeficiency, measured by CD4 lymphocyte serum count, and the value of ADA in the pleural fluid. Our research found significance differences between groups in ADA level. The AUC of the ROC curve achieved the threshold level of 0.75 that was reported as clinically useful. An high number of patients (88%) had confirmed bacteriologic disease.

The diagnostic utility of the ADA assay for pleural TB in HIV infected patients is still controversial. Hsu et al. found no statistical differences in the ADA assay between patients with immunodeficiency and TB diagnosis versus patients with pleural effusion of malignant etiology. They concluded that the diagnostic value of ADA in immunocompromised hosts with pleural tuberculosis effusions is not as significant as in immunocompetent hosts [5]. However, recent studies have showed a greater sensitivity and specificity for ADA assay, even in patients with low CD4 count [6,7]. In contrast, ADA assay showed a low sensitivity to diagnosis of TB in patients co-infected with HIV in other corporal fluids such as pericardial effusions and cerebrospinal fluid [10,11].

In conclusion, ADA level determination could be useful to diagnose pleural tuberculosis in HIV infected patients, although larger multicentric studies are necessary to confirm this. Conventional bacteriologic determinations are necessary to evaluate these patients in the meantime.

## Figures and Tables

**Figure 1 medsci-06-00101-f001:**
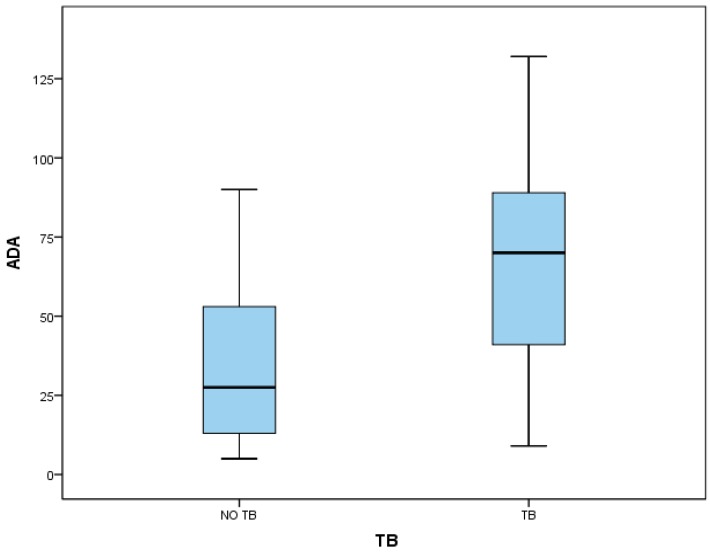
Box plot for ADA value according to final diagnosis. Abbreviations: ADA: adenosine deaminase assay; TB: diagnosis of tuberculosis; NO TB: diagnosis other than tuberculosis. In each box plot, the central horizontal line indicates the median value, and the lower and upper box horizontal lines indicate the 25th and 75th percentiles, respectively. Whiskers above and below the box indicate the 90th and 10th percentiles, respectively.

**Figure 2 medsci-06-00101-f002:**
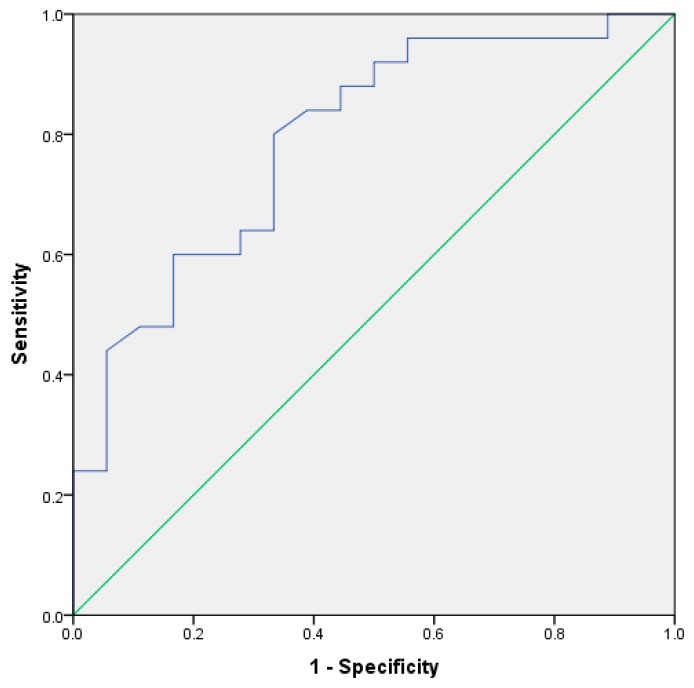
Receiver operating characteristic curve for adenosine deaminase assay to predict tuberculous pleural effusion.

**Table 1 medsci-06-00101-t001:** Patients characteristics. Data are median (IQR), mean (SD), or *n* (%).

Variables	TB (*n* = 25)	Non-TB (*n* = 18)	*p*-Value
Age	41 (±10)	40 (±10)	NS
Male sex	19 (76%)	13 (72%)	NS
ADA	70 IU/L (IQR 41–89)	27.5 IU/L (IQR 13.5–52)	NS
CD4	174 (IQR 86–274)	134 (IQR 71–371)	NS
Ph	7.41 (±0.13)	7.39 (±0.07)	NS
Glucose mg/dL	83 (±26)	66 (±27)	NS
Proteins g/dL	3.27 (±1.47)	3.95 (±1.15)	NS
LDH mmol/L	1046 ± 635	882 (±497)	NS
Leukocytes/mm^3^	1173 (±1733)	1375 (±1024)	NS

IQR: interquartile range; SD: standard deviation; TB: tuberculosis; ADA: adenosine deaminase; LDH: lactate dehydrogenase; NS: not significant; CD4: cluster of differentiation 4.

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
