# Peer review of "Usefulness of Adenosine Deaminase Assay in Diagnosis of Patients with HIV Infection and Pleural Tuberculosis"

_medsci, 2018, doi:10.3390/medsci6040101_

Round 1
Reviewer 1 Report
The authors investigated the usefulness of adenosine deaminase assay in diagnosis of patients with HIV infection and pleural tuberculosis. This is a very interesting topic and the methods are documented in sufficient detail however, one of the major concerns with the conclusions is the small size of population. I have several comments:
- There is no mention to how the sample size for the study was calculated. I would suggest that the authors clearly state how they reached the required sample size.
- The authors should stress whether several other studies failed to identify associations between CD4 lymphocytes count and the value of ADA.
- Please report if ADA analysis was performed blind to the diagnosis.
- Please prepare a table with the demographic characteristic of patients.
- Please improve the English language and style.
Author Response
- There is no mention to how the sample size for the study was calculated. I would suggest that the authors clearly state how they reached the required sample size.
* We take the patients who met the inclusion criteria within a certain time interval.
- The authors should stress whether several other studies failed to identify associations between CD4 lymphocytes count and the value of ADA.
* Accepted. we included this information in the manuscript, but we did not find more studies that analyze the relation between cd4 and ADA in HIV patients.
- Please report if ADA analysis was performed blind to the diagnosis.
* No. We think that is not necessary for a descriptive analysis.
- Please prepare a table with the demographic characteristic of patients.
* Accepted. We included a table in the new manuscript.
- Please improve the English language and style.
* Accepted.

Reviewer 2 Report
It was nice to read this study in what I feel is an important diagnostic area in HIV/TB.
This a reasonably sized patient group from one center but to understand the data clearly I feel more clinical characterization needs to occur.
In the abstract I think you do need to state CD4 counts of the study population as otherwise the reader cannot decide how meaningful the study will be to them.
In the results section I think it is crucial to know how many of the patients were on ARVs and how many had undetectable HIV loads. This is needed to understand lymphocyte function and background activation, and without it interpreting ADA is not possible. It also helps to inform what sort of TB presentations one might expect and what other illnesses might be causing pleural effusions.
I think how TB diagnoses were made has to be stated. I know most were culture-confirmed but what about the culture negatives and perhaps even more importantly what about those said not to have TB but without a clear alternative diagnosis? Did they have pleural biopsy? I would be interested to know what the ADA was in these. The study would be strengthened if it had clear diagnostic criteria to include or exclude TB.
The culture +ve rate is admirably high (88%). I think it would be useful to know not just how many were pleural fluid positive but also the other samples that were culture-positive.
Figure 2 - should the axes not meet at zero?
I was surprised that means with SD and not medians with ranges were presented. Could you please explain.
Author Response
This a reasonably sized patient group from one center but to understand the data clearly I feel more clinical characterization needs to occur.
* Accepted. we included a new table with more clinical characterization.
In the abstract I think you do need to state CD4 counts of the study population as otherwise the reader cannot decide how meaningful the study will be to them.
* Accepted. We included the cd4 counts in the abstract.
In the results section I think it is crucial to know how many of the patients were on ARVs and how many had undetectable HIV loads. This is needed to understand lymphocyte function and background activation, and without it interpreting ADA is not possible. It also helps to inform what sort of TB presentations one might expect and what other illnesses might be causing pleural effusions.
* we haven`t got this information complete. We have a lot of lost data, so we prefer no included it in the manuscript
I think how TB diagnoses were made has to be stated. I know most were culture-confirmed but what about the culture negatives and perhaps even more importantly what about those said not to have TB but without a clear alternative diagnosis? Did they have pleural biopsy? I would be interested to know what the ADA was in these. The study would be strengthened if it had clear diagnostic criteria to include or exclude TB.
*Accepted. We included this information in the results.
The culture +ve rate is admirably high (88%). I think it would be useful to know not just how many were pleural fluid positive but also the other samples that were culture-positive.
* It is not correct. In the text said .` Of 43 patients, 25 presented diagnosis of TB, 22 of them with positive culture (11 from pleural fluid and 11 from others respiratory samples)`
Figure 2 - should the axes not meet at zero?
* Accepted. we modify the figure
I was surprised that means with SD and not medians with ranges were presented. Could you please explain.
We change by the median with range in the data without Gaussian distribution like ada and cd4 count. And we used mean with SD in the others
